# Regulation of the Hypoxia-Inducible Factor (HIF) by Pro-Inflammatory Cytokines

**DOI:** 10.3390/cells10092340

**Published:** 2021-09-07

**Authors:** Mykyta I. Malkov, Chee Teik Lee, Cormac T. Taylor

**Affiliations:** 1Conway Institute of Biomolecular and Biomedical Research, University College Dublin, Belfield, Dublin 4, Ireland; mykyta.malkov@ucdconnect.ie (M.I.M.); chee.lee@ucdconnect.ie (C.T.L.); 2School of Medicine, University College Dublin, Belfield, Dublin 4, Ireland

**Keywords:** hypoxia, HIF, HIF-1α, inflammation, TNF-α, IL-1β

## Abstract

Hypoxia and inflammation are frequently co-incidental features of the tissue microenvironment in a wide range of inflammatory diseases. While the impact of hypoxia on inflammatory pathways in immune cells has been well characterized, less is known about how inflammatory stimuli such as cytokines impact upon the canonical hypoxia-inducible factor (HIF) pathway, the master regulator of the cellular response to hypoxia. In this review, we discuss what is known about the impact of two major pro-inflammatory cytokines, tumor necrosis factor-α (TNF-α) and interleukin-1β (IL-1β), on the regulation of HIF-dependent signaling at sites of inflammation. We report extensive evidence for these cytokines directly impacting upon HIF signaling through the regulation of HIF at transcriptional and post-translational levels. We conclude that multi-level crosstalk between inflammatory and hypoxic signaling pathways plays an important role in shaping the nature and degree of inflammation occurring at hypoxic sites.

## 1. Introduction

Eukaryotic cells generate metabolic energy currency within mitochondria in the form of adenosine triphosphate (ATP), utilizing molecular oxygen (O_2_) as the final electron acceptor in the oxidative metabolism of glucose (oxidative phosphorylation) [1]. Cells therefore require a continuous O_2_ supply in order to maintain bioenergetic homeostasis. The O_2_ required for this is provided by the combined activity of the pulmonary, hematopoietic, and cardiovascular systems which transport oxygen from the atmosphere to individual cells. Tissue oxygenation homeostasis is therefore determined by a balance of O_2_ consumption by mitochondria and oxygen supply by erythrocytes in capillaries [2]. Hypoxia is the condition which arises when the cellular O_2_ demand required to generate sufficient levels of ATP to support physiological requirements exceeds the available supply. This can be as a result of an increased O_2_ demand and/or decreased supply to the tissue.

Despite its inherent challenge to homeostasis, hypoxia is frequently encountered and associated with physiological conditions including fetal development, adaptation to altitude, and physical exertion. For example, climbers who ascend to 8400 m at the summit of Mount Everest experience severe hypoxemia and their alveolar-arterial oxygen differences are elevated [3]. Hypoxia can also occur in a range of pathological diseases including inflammation, cardiovascular disease, and cancer, where oxygen demand is enhanced or oxygen supply is diminished [4]. For example, in the context of coronary artery disease, the decrease in myocardial perfusion is caused by the formation of atherosclerotic plaques in the main coronary arteries, consequently creating tissue hypoxia [5]. Therefore, hypoxia is a commonly encountered stress in both health and disease which can promote physiologic or pathological responses.

## 2. Hypoxia and Inflammation

Tissue oxygen levels within the body are lower than in the atmosphere (21% O_2_). Tissues carry out their physiological functions between 15% O_2_ (≈100 mmHg) and 1% O_2_ (≈7 mmHg) [6]. The anatomical localization of a tissue also determines physiological oxygen gradients within the body. For example, the gastrointestinal tract is characterized by a unique oxygen profile in which a steep trans-mucosal oxygen gradient across the epithelial layer exists. A continuous state of physiologic hypoxia within gastrointestinal tissue occurs because firstly, the intestinal epithelial cells are located between the anoxic lumen of the gut and the highly vascularized mucosal capillary bed, and secondly, the blood perfusion of the intestinal mucosa fluctuates throughout the day depending on digestion [7].

Many clinical studies have demonstrated that hypoxia is a prominent microenvironmental feature of inflammatory pathologies including atherosclerosis, cancer, and inflammatory bowel disease (IBD) [8]. In atherosclerosis, increases in the thickness of the arterial wall leads to hypoxia within the intima which subsequently decreases tissue perfusion and thereby promotes the development of proatherosclerotic processes [9]. Impaired oxygen consumption and delivery in solid tumors arises from the limited oxygen diffusion in avascular primary tumors and enhanced oxygen consumption caused by hyperproliferating cancer cells [10]. The presence of hypoxia restricted to the epithelial surface in animal models of IBD has been observed by using 2-nitroimidazole dyes [11]. Using endoscopic oximetry, prominent levels of mucosal hypoxia correlated with the degree of inflammation were found in ulcerative colitis (UC) patients [12]. Contributors to tissue hypoxia in inflamed tissues include increased oxygen consumption by infiltrating immune cells which utilize O_2_ to sustain synthesis of inflammatory cytokines and mediators, as well as impaired oxygen supply to these tissues caused by vascular damage, thrombosis, and edema [13].

Hypoxia stimulates the expression of pro- and anti-inflammatory cytokines in different cell types and tissues of high-altitude residents [14]. For example, after spending three consecutive nights above 3400 m, healthy volunteers demonstrate elevated levels of circulating IL-6, IL-6 receptor, and C-reactive protein [15]. In addition, recent studies have shown elevated serum levels of IL-1β, IL-6, IL-8, IL-10, and TNF-α in healthy volunteers and volunteers with acute mountain sickness (AMS) who ascended to 3800 m [16,17]. In summary, as illustrated in Figure 1, extensive evidence now suggests that hypoxia is a prominent feature of the inflammatory microenvironment in multiple conditions, thereby emphasizing the importance of investigating hypoxic mechanisms involved in the context of inflammation.

## 3. The Hypoxia-Inducible Factor (HIF)

Due to the essential role of molecular oxygen in the maintenance of biological activity of eukaryotic cells and the common occurrence of hypoxia, metazoan organisms have evolved molecular mechanisms to sense and quickly respond to changes in oxygen levels by initiating adaptive responses [18]. Importantly, despite our knowledge of physiological responses to hypoxia, the mechanistic understanding of how cells sense and adapt to hypoxia remained unclear until the discoveries of the 2019 Nobel Prize winners in Medicine and Physiology—Gregg Semenza, William Kaelin, and Peter Ratcliffe. On a cellular level, tissue adaptation to hypoxia is largely controlled by the hypoxia-inducible factor (HIF) signaling pathway. HIFs are heterodimeric transcription factors ubiquitously expressed by all metazoans which regulate the cellular response to decreased oxygen availability through the expression of hundreds of hypoxia-dependent genes [19]. These genes are involved in processes such as angiogenesis, energy metabolism, erythropoiesis, and cell survival, consequently promoting oxygen homeostasis in hypoxic tissues [20]. HIF is composed of one of three oxygen-dependent alpha subunits (HIF-1α, HIF-2α, and HIF-3α) and a constitutively expressed oxygen-insensitive beta subunit (HIF-1β/ARNT). Both HIF-1α and HIF-2α share similar structure and are associated with increased gene expression in hypoxia; however, they regulate overlapping yet distinct sets of target genes depending on oxygen availability and tissue type [21]. For instance, HIF-1α is ubiquitously expressed, whereas HIF-2α and HIF-3α are expressed within specific cells, such as lung type II pneumocytes, kidney interstitial cells, and liver parenchymal cells [22]. The transcriptional response to hypoxia is orchestrated through the activation of the HIF pathway.

Under conditions of normoxia, when cellular oxygen supply exceeds demand, the majority of molecular oxygen delivered to a cell is used to generate ATP through mitochondrial oxidative phosphorylation, thus satisfying cellular metabolic demand [23]. Some of the remaining oxygen facilitates the activity of a family of Fe^2+^ and 2-oxoglutarate-dependent (2-OG) dioxygenases termed prolyl hydroxylase domain (PHD) enzymes (also called egl nine homolog (EGLN)) including PHD1/EGLN2, PHD2/EGLN1, PHD3/EGLN3, and a single asparagine hydroxylase called factor-inhibiting HIF (FIH) [24]. PHD activity is dependent on molecular oxygen, therefore, they have been suggested to be physiological molecular oxygen sensors [20]. Moreover, PHD2 is the primary HIF hydroxylase in most cell types as it confers most PHD activity in normoxia [25]. PHDs hydroxylate HIFα subunits in an O_2_-dependent manner at two highly conserved target proline residues in the NH_2_-terminal oxygen-dependent degradation domain (NODDD, Pro402 on HIF-1α) and the COOH-terminal oxygen degradation domain (CODD, Pro564 on HIF-1α) utilizing molecular oxygen, Fe^2+^, and 2-OG as co-factors [24,26]. The HIF pathway can be further modulated by transcription factors such as Nuclear Factor-kappa B (NF-κB), microRNAs (miRNAs), long noncoding RNAs (LncRNAs), and multiple post-translational modifications (e.g., SUMOylation, acetylation, methylation, and phosphorylation) [27,28,29]. Hydroxylation of the proline residues target HIF-α subunits for ubiquitination, which is mediated by the E3 ubiquitin ligase complex named the von Hippel-Lindau (VHL) protein [30]. This process leads to ubiquitin-dependent proteasomal degradation of HIF. In addition, FIH regulates HIF transcriptional activity by hydroxylating an asparagine residue in the C-terminal transactivation domain (Asn803 on HIF-1α), thereby preventing binding of HIF with the transcriptional co-activator proteins, CREB-binding protein (CBP) and p300 [31]. Thus, in normoxia, PHDs and FIH promote a dual mechanism of HIF degradation, leading to suppression of HIF transcriptional activity (Figure 2).

Under conditions of hypoxia, virtually all of the molecular oxygen available is consumed during mitochondrial oxidative phosphorylation, thereby inhibiting the catalytic activity of the PHDs and FIH [32]. This process limits hydroxylation of HIF-α subunits and subsequent HIF degradation, thus activating the HIF pathway. Stabilized HIF-α subunits translocate to the nucleus where they dimerize with HIF-1β subunits, thereby forming transcriptionally active HIF-αβ heterodimers [20]. HIF-αβ heterodimers are then free to bind to hypoxia-response elements (HREs) within the enhancers or promoters of HIF target genes and recruit the transcriptional co-activator CBP/p300 [33,34]. This functionally active transcriptional complex promotes regulation and expression of HIF-dependent adaptive genes. For instance, activation of the HIF pathway in cardiomyocytes and endothelial cells leads to enhanced expression of the downstream genes including vascular endothelial growth factor (VEGF), endothelin-1(ET-1), and inducible nitric oxide synthase (iNOS), consequently, increasing angiogenesis [35]. Hypoxia-induced activation of HIF as well as pharmacologic inhibition of HIF-hydroxylases induces transcription of genes including erythropoietin (EPO) and its receptor transferrin in vivo and in vitro, thereby ameliorating anemia [36,37]. Moreover, HIF-dependent genes include a number of glycolytic enzymes and genes such as pyruvate dehydrogenase kinase 1 (PDK1), which are capable of decreasing the basal respiratory rate [19]. Therefore, the coordinated activation of HIF-dependent target genes in response to decreased oxygen availability subsequently leads to increased cellular oxygen delivery.

## 4. The Involvement of HIF in Cytokine Production

The importance of HIF-1α in immunity has been described previously [38,39,40]. During inflammation, the metabolic demand of immune cells required to produce inflammatory cytokines, enzymes, and inflammatory mediators is dramatically increased as cells migrate from the well-oxygenated vasculature to hypoxic inflamed regions; therefore, evidence is mounting that suggests that HIF exerts a pivotal role in the adaptation to such conditions. The HIF pathway has a strong impact on epithelial and immune cell function and development during inflammation via the activation of adaptive responses in these cells [41]. HIF-1α is expressed in all immune cells, whereas the expression pattern of HIF-2α is limited to several sub-types such as T cells, tumor-associated macrophages, and neutrophils [40]. HIF-1α has been reported to directly regulate the production of pro-inflammatory cytokines TNF-α, IL-1β, IL-6, and IL-8 in rheumatoid arthritis synovial fibroblasts (RASF), thus, mediating interactions between T-cell/B-cell and RASF [42]. Another recent study demonstrated that in a mouse model of dextran sulfate sodium (DSS) induced colitis, the absence of dendritic HIF-1α results in enhanced severity of intestinal inflammation via increased production of pro-inflammatory cytokines IL-6 and IL-23 [43]. HIF-1α and HIF-2α exert different functions in the intestinal epithelium and other tissues; therefore, the balance of HIF-1α versus HIF-2α has important consequences for inflammation. Mice with constitutive HIF-1α expression due to VHL protein inhibition demonstrated elevated levels of TNF-α, IL-1β, and IL-6 which in turn promote increased inflammatory infiltrates and colonic oedema [44]. Interestingly, HIF-2α deletion in a mouse model of moderate DSS-induced colitis was shown to be protective, whereas genetic HIF-2α overexpression in intestinal epithelial cells resulted in spontaneous DSS colitis via increased expression of TNF-α, IL-1β, and IL-6 [45]. HIF-2α also plays a distinct role as transcriptional regulator of TNF-α, IL-1β, IL-6, and IL-12 expression in macrophages via direct binding to the promoters of cytokine genes under conditions of hypoxia [13]. Therefore, the HIF pathway plays a direct role in the regulation of cytokine production by multiple cell types under normoxic and hypoxic conditions.

Given the importance of the PHDs and FIH in the regulation of the HIF pathway, pharmacologic manipulation of these enzymes became an area of active research with respect to investigating new therapeutics for treatment of inflammatory disorders. The pharmacologic pan-hydroxylase inhibitor dimethyloxalylglycine (DMOG) is a widely used 2-OG mimetic which suppresses the activity of both proline and asparaginyl hydroxylases in vivo and in vitro [46]. For instance, in 2008, Cummins et al. [47] demonstrated for the first time that activation of the HIF pathway by mimicking hypoxia through pharmacologic hydroxylase inhibition significantly reduced disease severity and levels of inflammatory markers IL-1β, IL-6, IL-12, and TNF-α via stabilization of HIF-1α in a mouse model of DSS-induced colitis [47]. It was recently shown that activation of HIF-1α by DMOG plays a protective role in the development of apical periodontitis via suppression of proinflammatory cytokines TNF-α and IL-1α in macrophages in vitro [48]. In addition, DMOG was shown to downregulate IL-1β, IL-6, IL-8, and TNF-α gene expression in human gingival fibroblasts stimulated with *Fusobacterium nucleatum*, which is widely used to induce inflammation in experimental periodontitis models [49]. These studies highlight the important anti-inflammatory properties of hydroxylase inhibitors in different models of inflammation. Based on the evidence from multiple studies outlined above, inflammatory hypoxia promotes protective transcriptional responses mediated by HIF pathway thereby marking HIF as a key mediator of adaptive and innate immune response during inflammation [50]. HIF may be considered as a potentially useful therapeutic agent for the treatment of inflammatory disorders and, therefore, further elucidation of the role of inflammatory cytokines as a potential mediator of the HIF pathway could develop novel therapeutic strategies to treat inflammatory disorders.

## 5. Regulation of HIF by Cytokines

As outlined above, inflammatory disorders are frequently characterized by microenvironmental hypoxia with activation of the HIF pathway. Conversely, pathological conditions that are caused by hypoxia are characterized by secondary inflammatory changes [8]. Therefore, the crosstalk between inflammatory cytokines and hypoxia has important implications for inflammation. While the growing body of evidence generated over the past years highlights the association of hypoxia an oxygen-sensing pathways with inflammation as well as potential therapeutic intervention of the HIF pathway in different inflammatory models, there is still limited knowledge relating to how inflammatory cytokines affect hypoxia-dependent mechanisms and regulation of the HIF pathway. Inflammation may alter the machinery involved in the activation of the HIF pathway, thereby changing sensitivity of the tissue to hypoxia even before hypoxia is present in inflammatory foci. In addition, the presence of cytokines can combine with hypoxia to regulate immune system function which, in turn, will regulate the protective effects of hypoxia [41]. Therefore, it has been proposed that the presence of a rich pro-inflammatory cytokine network in tissue microenvironment plays a central role in regulating the HIF pathway. Amongst some of the well-studied growth factors, chemokines, and cytokines, TNF-α and IL-1β play a central role in inflammation that has been recognized as a key step in angiogenesis, tumor survival, and local invasiveness [39,51]. In this review we focused on the role of TNF-α and IL-1β in the regulation of the HIF pathway.

### 5.1. Regulation of HIF by TNF-α

TNF-α is a major pro-inflammatory cytokine produced by many cell types including (but not limited to) macrophages, natural killer cells, and T lymphocytes and it plays a crucial role in regulating inflammatory effects and host defense against microbial pathogens in a broad range of tissues and cell types [52]. While most studies have focused on hypoxia-dependent regulation of the HIF pathway, HIF activation independent of hypoxia has been less well understood. Normoxic regulation of HIF by inflammatory cytokines was firstly described by Hellwig-Burgel et al. [53], where they demonstrated that treatment of human hepatoma (HepG2) cells with TNF-α resulted in elevated HIF-1 activity and HIF-1 DNA binding; however, neither HIF-1α mRNA nor protein levels were reported to be affected by TNF-α, thus implicating a post-translational mechanism [53]. Regulation of HIF activity by TNF-α can also occur via a variety of common intracellular signaling pathways, including NF-κB and phosphatidylinositol-3 kinase (PI3K)/protein kinase B(Akt) pathways. TNF-α is known to activate PI3K/Akt pathway, which is essential in the accumulation of HIF-1α in response to hypoxia in HepG2 cells [54]. Moreover, it was recently shown that TNF-α dependent NF-κB pathway activation upregulates HIF-1α mRNA and protein levels after 24-h treatment in human pterygium fibroblasts (HPFs) independent of hypoxia [55]. In line with these findings, TNF-α induces VHL-HIF-1α interaction and increases HIF-1α accumulation and ubiquitination through an NF-κB-dependent pathway in human embryonic kidney (HEK293) cells under normoxic conditions [56].

It was proposed that stimulation with TNF-α activates NF-κB via phosphorylation of inhibitory κB (IκB) which then binds to consensus site in the HIF-1α promoter, thus elevating HIF-1α protein and mRNA levels [57]. A subsequent study demonstrated that HIF-1α is transcriptionally induced by the action of TNF-α in HEK293 cells through the NF-κB pathway as described above [58]. Treatment of human bronchial airway smooth muscle cells (ASMCs) with TNF-α promoted p65 NF-κB phosphorylation and nuclear accumulation, thus enhancing HIF-1α protein and mRNA levels under both normoxia and hypoxia [59]. Interestingly, the same study demonstrated that ex vivo treatment of rabbit tracheal strips with TNF-α resulted in downregulated HIF-1 activity caused by diminished dimerization of HIF-1α with HIF-1β and, therefore, reduced binding of HIF-αβ heterodimers to HREs under hypoxia [59]. While there exists limited information relating to the role of HIF-3α in inflammation, TNF-α was reported to increase HIF-3α mRNA expression in rat PC12 cells and NF-κB regulate HIF-3α mRNA expression in human umbilical venous endothelial cells [60,61].

During acute inflammation, TNF-α can induce HIF-1α expression in inflammatory macrophages harvested from wounds, and maintained under normoxic conditions, consequently, the expression pattern of VEGF and iNOS, which are key HIF-dependent genes involved in the wound healing, can be affected in response to TNF-α treatment [62]. Moreover, stimulation of murine skeletal muscle myocytes with TNF-α resulted in a dose-dependent increase in expression of pVHL and ubiquitin-conjugating enzyme 2D1 (Ube2D1), which are known to regulate muscle angiogenesis through the regulation of the HIF pathway. TNF-α treatment also increased PHD2 protein expression and total cellular ubiquitin levels [63]. However, TNF-α treatment reduced HIF-1α protein expression in skeletal muscle myocytes under normoxic conditions [63]. It was recently shown that under normoxic conditions stimulation of human intestinal epithelial Caco-2 cells with pro-inflammatory cytokines including TNF-α downregulated PHD1 mRNA levels which were correlated with decreased CCAAT-enhancer binding protein (C/EBPα) mRNA and protein expression, suggesting that the effect of TNF-α on the C/EBPα/PHD1 mechanism could lead to increased HIF-1α stabilization in the mucosal tissue of UC patients [12]. TNF-α treatment also significantly upregulates HIF-1α protein and mRNA levels in human squamous lung A549 and H226 cells, which inhibits A549 cell proliferation and adhesion as well as TNF-α-activated transcriptional activity of HIF-1α, resulting in suppression of vasodilator-stimulated phosphoprotein (VASP) and consequent inhibition of transplanted tumors growth in nude mice [64]. Taken together, TNF-α can be considered a modulator of the sensitivity of the HIF pathway to hypoxia in inflamed cells and its regulatory and constituent components, thus modulating protective HIF responses in different cell types and tissues. The effects of TNF-α on HIF-1α are summarized in Figure 3.

### 5.2. Regulation of HIF by IL-1β

IL-1β is a pro-inflammatory cytokine secreted mainly by monocytes and macrophages, characterized by its diverse physiological and pathological functions [65]. HIF-1α protein expression is induced by IL-1β in normoxia in different cell types including normal human cytotrophoblast (CTB) cells, hepatocytes, human osteoarthritic (HC-OA) chondrocytes, gingival fibroblasts (HGFs), synovial fibroblasts (HSFs), and an invasive breast cancer cell line MDA-MB-231 [54,66,67,68,69]. The accumulation of the HIF-1α protein was detected as early as 4 hours after initiation of treatment with IL-1β and correlated with the concentration and exposure time with treatment of IL-1β.

The effect of IL-1β on HIF-1α mRNA expression appears to be somewhat cell type-specific. In HGFs and HSFs, treatment with IL-1β increased HIF-1α mRNA in a time-dependent manner [68]. In contrast, IL-1β did not induce HIF-1α mRNA expression in human cytotrophoblast cells and HepG2 cells, demonstrating cell-type specific effects of these agents on HIF-1α protein expression [54,67]. While increased HIF-1 DNA-binding was observed in HGFs, HSFs, and HepG2 cell lines after they were treated with IL-1β, a reporter assay was used to determine if IL-1β affects HIF-1α-dependent gene expression. While IL-1β had no effect on reporter gene expression in normoxia, co-treatment of HepG2 cells with hypoxia and IL-1β amplified HIF-1 reporter gene activity by 25% compared with hypoxia alone [54].

VEGF protein secretion in normal human cytotrophoblast cells is stimulated by IL-1β in a dose-dependent manner. IL-1β levels as low as 1 ng/mL increased VEGF protein expression by normal human cytotrophoblasts [67]. In MDA-MB-231 cells, IL-1β treatment upregulated VEGF and cell surface CXC chemokine receptor 1 (CXCR1) mRNA expression. Of note, the transcription of CXCR1 promotes cell survival and is specifically mediated by HIF-1α [70]. CXCR1 mediates the biological effects of chemokine CXCL8 and is actively involved in angiogenesis and cell migration. Synergistically, these results suggest that IL-1β induces HIF-1-responsive gene expression under normoxia in MDA-MB-231 cells. CXCL8 associated with cell motility has been identified in breast cancer stem cells, which highlights its relevance in cancer progression [71].

While previous studies have suggested that the intermediatory role of IL-1β involves biological activation of the mitogen-activated protein kinase (MAPK) signaling pathway, Stiehl et al. [54] described the downregulatory effect of MAPKKs inhibitors, PD 98059 and U0126, on IL-1β-induced HIF-1α accumulation and HIF-1 DNA-binding in HepG2 cells [54]. The other finding reported by the same group was that LY 294002, a PI3K inhibitor, suppressed HIF-1α activation in a dose-dependent manner despite the treatment with IL-1β. This inhibitory effect was further illustrated when the production of erythropoietin was fully blocked and that of vascular endothelial growth factor reduced following inhibition of the PI3K pathway [54]. Meanwhile, the treatment with IL-1β and LY 29400 significantly downregulated HIF-1α gene expression in osteoarthritic chondrocytes [69]. Protein phosphatase 2A (PP2A) has also been studied and its inhibition, knock-down or knock-out, enhances the activation of MAPKs in response to pro-inflammatory stimuli, leading to increased expression of inflammatory mediators [72].

While evidence suggests that HIF-1α transcriptional activity is modulated by phosphorylation, which p42/44 (extracellular signal-regulated kinase ERK 1/2) MAPKs was known to catalyze the phosphorylation thereof, it remained a question whether IL-1β-regulated-HIF-1α expression in normoxic cells involves ERK1/2 signaling until Qian and co-workers reported their findings [67,73,74]. Using antibodies that specifically bind the phosphorylated form of ERK1/2, 10 ng/mL of exogenous IL-1β was found to increase the phosphorylation of ERK1/2 [67]. However, pre-treatment of the normal human cytotrophoblast cells with PD98059, which inhibits ERK1/2, blocked the induction of HIF-1α protein expression in IL-1β-treated cells in a dose-dependent fashion. PD98059 was also found to inhibit the induction of VEGF protein secretion [67]. These findings reinforce that the positive regulation of HIF-1α by IL-1β occurs at least in part post-transcriptionally. The hypothesis that the regulation of HIF-1α by IL-1β involves the PI-3K pathway is supported, since the cells treated with inhibitors of this pathway displayed no effects of IL-1β. Post-transcriptional regulation of HIF-1α by IL-1β may suggest that elevated protein levels increase the possibility of the binding of HIF-1α to its response element and the subsequent gene activation that is crucial to overall energy expenditure and cell survival.

Translationally, this effect was observed in tumor cell migration. Naldini et al. [66] observed that non-hypoxic induction of HIF-1α by IL-1β in MDA-MB-231 was found to be associated with increased tumor cell migration, coupled with induction of p38 MAPK phosphorylation and CXCL8/CXCR1 expression [66]. Using siRNA to inhibit HIF-1α resulted in a significant decrease in CXCR1 expression and IL-1β-induced cell migration in MDA-MB-231 cells, which confirms a role of HIF-1α in the non-hypoxic-IL-1β-dependent induction of migration. This observation was further illustrated in tumor cells growing in vivo (nude mice) for 3 weeks, trying to mimic the endogenous release of IL-1β in mice bearing MDA-MB-231 xenografts. When compared with tumor specimens from mice bearing colony not secreting IL-1β, MDA-MB-231 xenografts harvested from mice bearing colony secreting IL-1β exhibited a higher protein expression of HIF-1α [66]. These findings support the link between inflammation and cancer. In summary, evidence supports the model that IL-1β regulates HIF-1α protein post-transcriptionally, subsequently increasing the HIF-1α functional response in cells (Figure 3).

## 6. HIF and Cytokines Crosstalk in Cancer and Inflammation

Chronic inflammation can promote metaplasia, dysplasia, and eventually neoplasia [75,76,77,78,79]. The fundamental roles of how primary tumor microenvironment (TME) contributes to cancer growth, progression, invasion, and metastasis have been intensively studied over the last decade [80,81]. This intricate network of crosstalk between epithelial cancer cells, tumor-educated stromal cells including mesenchymal stem cells (MSCs) and cancer-associated fibroblasts (CAFs), infiltrating immune cells like macrophages, lymphocytes, and granulocytes enrich the TME with pro-inflammatory cytokines and chemokines [82,83,84,85]. Two of these pro-inflammatory cytokines are TNF-α and IL-1β. Chronic exposure to TNF-α and IL-1β leads to tumor progression as these cytokines are pro-angiogenic and promote epithelial-mesenchymal transition (EMT) and cell migration [86,87,88]. TNF-α and IL-1β drive these downstream effects through HIF-1α protein stabilization. Furthermore, HIF-1α and NF-κB work synergistically to regulate the transcription of hundreds of genes that, in turn, control vital cellular processes such as metabolic adaptation and reprograming, inflammatory repair response and extracellular matrix digestion [83,89,90,91,92,93]. The key step that integrates hypoxic adaptation with inflammatory reparation is the discovery of alarmin receptors genes such as RAGE and P2X7. These alarmin receptors are activated by HIF-1α and strongly induce NF-κB and proinflammatory gene expression which, subsequently, lead to the acquisition of crucial properties of the malignant tissue phenotype [91]. This includes pro-tumorigenic chemokines CCL2, CCL5, and CXCR1/CXCL8, which are the key initial steps in cell migration in different cell types, including triple-negative breast cancer cells [94,95,96,97,98]. Findings of higher levels of TNF-α plasma levels in various malignancies such as gastrointestinal carcinoma than in controls further support this observation [99,100].

Moreover, in the Canakinumab Anti-Inflammatory Thrombosis Outcomes Study (CANTOS), treatment with anti-IL-1β (Canakinumab) was reported with lower incidence of lung cancer by 67% and mortality by 77% compared with the placebo group. It had been proposed that anti-IL-1β treatment decreased the inflammation that would prove significant in progression to tumor development [101,102,103]. Trials that added IL-1β receptor antagonist (Anakinra) to standard-of-care chemotherapy for colorectal cancer and pancreatic ductal adenocarcinoma showed significant survival profiles [104]. Inhibitors of TNF-α and IL-1β that are used in inflammatory diseases were also found to inhibit the aggressive phenotype of triple-negative breast cancer [105,106]. In summary, TNF-α and IL-1β play a central role in inflammation-mediated carcinogenesis. The recent implementation of inhibitory modalities to these cytokines on top of standard-of-care therapy has proved successful in inhibiting inflammation-mediated immunosuppression and invasiveness in tumor progression [104,105,106]. The possibility of blocking these cytokines as an adjunctive treatment needs to be further studied in vivo and even more so in the clinical setting in order to improve our understanding in its effect in different contexts such as tumor origin, staging, and grading.

The key role of TNF-α in pathogenesis of IBD has also been widely discussed [107,108,109]. IBD is characterized by a loss of intestinal epithelial barrier function with subsequent unregulated leakage of gut microbiota and antigenic material from the lumen into the intestinal submucosal tissue, thereby promoting inflammation which further drives barrier loss and disease progression [110]. Despite the high global prevalence of IBD, therapeutic management is still insufficient. Nevertheless, since the introduction of anti-TNF-α therapy in the treatment of IBD in 1998, TNF-α is now considered as a major cytokine in this disease [111,112]. Previous clinical studies have demonstrated that inhibition of TNF-α inflammatory pathway by using anti-TNF-α monoclonal antibodies, such as infliximab, promoted positive outcomes in IBD patients; however, only two-thirds of the patients improved after such therapy [113,114]. Moreover, anti-inflammatory drugs such as 5-Aminosalicylates are frequently used for anti-inflammatory therapy of patients with UC, whereas in patients with Crohn’s disease (CD), administration of these drugs demonstrated little or no efficacy in resolving of tissue inflammation [115]. To date, around 30% of patients with IBD do not respond to anti-TNF-α treatment and 50% of patients who developed an initial response to such therapy lost responsiveness to it within one year post-therapy [116]. Therefore, improved therapeutic strategies in this field are a clear clinical need. As previously discussed, multiple studies now have demonstrated that pharmacologic hydroxylase inhibition is protective in models of experimental colitis [47,48,117,118,119,120,121]. The evidence outlined in these studies strongly indicates that hydroxylase inhibitors exhibit HIF-dependent protective mechanisms that contribute to intestinal epithelial barrier protection, thus emphasizing their therapeutic potential in the treatment of IBD. It is important to note that the effects of hydroxylase inhibitor AKB-4924 have recently been examined in healthy male volunteers in phase Ia clinical trials [122,123]. Taking into consideration the upregulatory effects of TNF-α on the sensitivity of the HIF pathway and its regulatory components in inflamed tissues, in theory, inhibition of TNF-α in patients with IBD can lead to downregulation of HIF-1α and its consequent protective effects on barrier function, thus suggesting a potential reason for unresponsiveness of a large number of patients to such therapy. It was recently demonstrated that inhibition of TNF-α in patients with IBD resulted in the downregulation of HIF-1α mRNA levels [124]. Therefore, the crosstalk between anti-TNF-α therapy and pharmacologic activation of the HIF pathway has important outcomes for development of novel therapeutic strategies to treat inflammatory disorders and needs to be further studied.

## 7. Conclusions

To conclude, hypoxia and inflammation are co-incidental events in multiple physiological and pathological conditions, thereby emphasizing the importance of discussing the crosstalk between them. Importantly, the cellular adaptation to hypoxia is controlled by the HIF pathway which also plays a major role in the regulation of immune responses in inflammation. The HIF pathway directly promotes production of pro-inflammatory cytokines by many cell types both under hypoxia and independent of it. Even though significant work has been carried out towards investigating the protective mechanisms of HIF in inflammatory conditions, the mechanism by which cytokines regulate the HIF pathway is less characterized. Here, we summarized that two major pro-inflammatory cytokines, TNF-α and IL-1β, affect the sensitivity of the HIF pathway in various models of inflamed cells. TNF-α regulates transcriptional levels of HIF-1α via the NF-κB pathway; however, the mechanisms by which TNF-α directly affects HIF-1α post-transcriptionally/translationally remains unknown. On the other hand, the effects of IL-1β on HIF-1α occur post-transcriptionally depending on the cell type. While TNF-α and IL-1β modulate the HIF pathway at different levels, the combined effects of these two cytokines, as would normally be seen in pathophysiological inflammatory conditions, might result in dramatic shifts in the sensitivity of the HIF pathway. Little is known about how cytokines affect other intermediate transcriptional factors such as miRNAs and LncRNAs, as well as post-translational modifications including SUMOylation. Therefore, the impact of TNF-α and IL-1β on post-transcriptional and post-translational levels of HIF requires further research.

## Figures and Tables

**Figure 1 cells-10-02340-f001:**
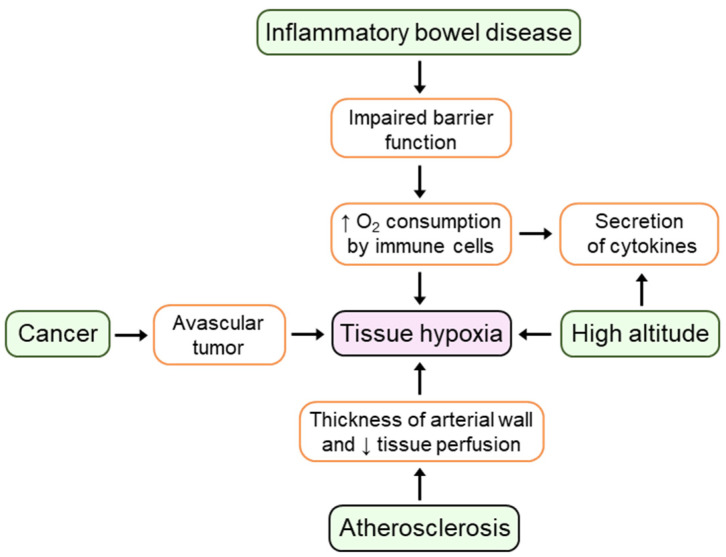
The presence of tissue hypoxia is common in physiology and disease.

**Figure 2 cells-10-02340-f002:**
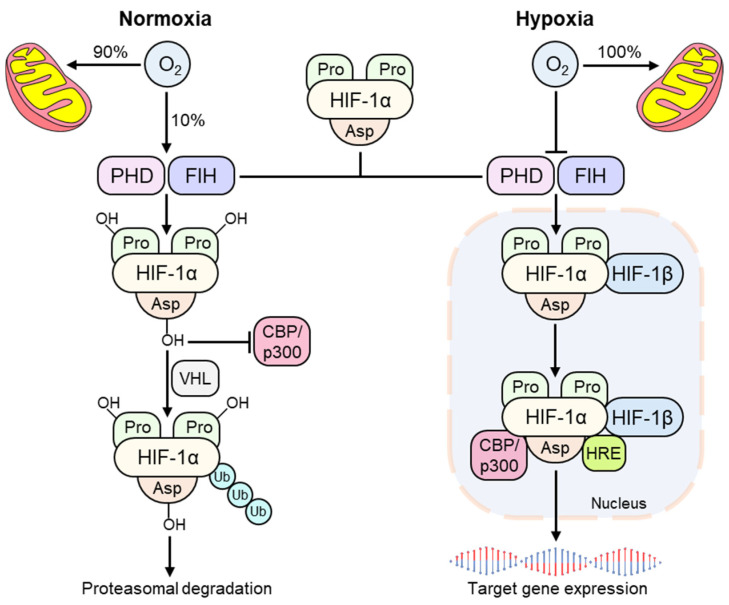
The hypoxia-inducible factor (HIF) pathway. In normoxia, most O_2_ is consumed by the mitochondria and remaining O_2_ facilitates hydroxylation of HIF by PHDs and FIH, leading to ubiquitin attachment by VHL E3 ubiquitin ligase. This results in proteasomal degradation and transcriptional repression of HIF. In hypoxia, virtually all O_2_ is used by the mitochondria resulting in hydroxylase inhibition. Stabilized HIF-α dimerizes with HIF-1β thereby forming a heterodimeric complex which binds to HRE and CBP/p300, leading to increased transcriptional expression of HIF-dependent genes.

**Figure 3 cells-10-02340-f003:**
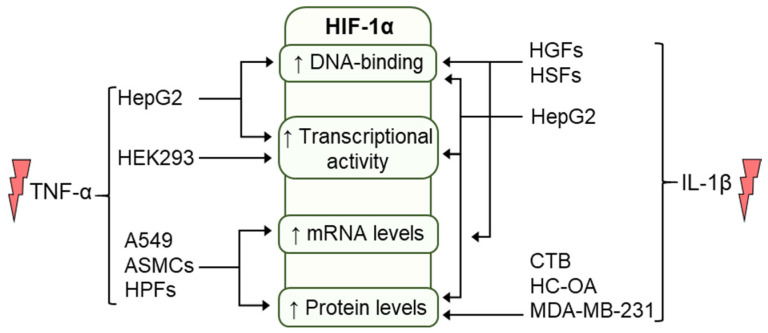
Effects of TNF-α and IL-1β stimulation on HIF-1α DNA-binding, transcriptional activity, mRNA, and protein levels in different cell types.

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
