# Peer review of "Regulation of the Hypoxia-Inducible Factor (HIF) by Pro-Inflammatory Cytokines"

_cells, 2021, doi:10.3390/cells10092340_

Round 1

Reviewer 1 Report

The paper by Malkov et al. “Regulation of the hypoxia-inducible factor (HIF) by pro-inflammatory cytokines” the authors give an interesting overview of the recent literature of the impact of inflammatory stimuli on HIF pathway with a focus on the two major pro-inflammatory cytokines TNF-α and IL-1β.

In the last years several studies dealing with the hypoxia as a modulator of immune responses were published and HIF as a potential therapeutic target for treatment of inflammatory disorders is described. The authors placed, as already mentioned, a special focus on TNF-α and IL-1β without addressing the origin of these cytokines in detail.

All in all, the review is well written and gives an overview of the IL1-β and TNF-α induced HIF activity in vitro, but leaving open the question if the cytokines boost the HIF activity or if HIF is responsible for the cytokine secretion during inflammation. This question can probably not be answered until now.

Just a few comments where the manuscript could be changed:

The authors mention 3 HIFa subunits but only HIF1a and HIF2a are discussed further. Are there any results for HIF3a in inflammatory diseases?

Figure 3 summarizes the effects of TNF-α and IL-1β on HIF1. The authors also describe an effect of this two cytokines on HIF regulating proteins (pVHL, PHD2) and HIF target genes. Maybe these can also be included into figure 3.

The authors state, that pharmalogic hydroxylase inhibition by AKB-4924 is examined in IBD patients. Reference 118 states that with the help of healthy male volunteers (and not IBD patients as written in the text) the safety and tolerability is tested. Results of further studies are published (doi.org/10.1093/ibd/zaa010.023) and should be mentioned.

As the authors state in their last part of the paper, anti-TNF-α therapy is part of the IBD treatment. They speculate that anti-TNF- α treatment can lead to downregulation of HIF-1 α, which is already described in literature, e.g. by Sun et al (2016). Furthermore, a combination of anti-TNF-α therapy and a pharmacological activation of the HIF pathway is suggested as a potential novel therapeutic strategy to treat inflammatory disorders. Although, inhibition of hydroxylases is under intensive research, until now the data on the role of the different hydroxylases in different cell types and their effect on intestinal inflammation are wide-ranging (see Van Welden, 2017).

Reviewer 2 Report

This concept has been suggested by many studies over the past 20 years and a focussed review is therefore worth attention. Several physiological and biochemical concepts have been described already. One question intrigues me: what is reported in literature and what is the opinion of these authors on the involvement of protein phosphatases/PP2A in regulating hypoxia, cytokine response and the regulation of transcription? The interplay between several cellular processes, especially cell survival and proliferation, these phosphatases regulate protein kinases and subsequent down-stream processes. This aspect is expected to become more important in the future and a possible drug target in disease.
